

# Multi-sensor network tracking research utilizing searchable encryption algorithm in the cloud computing environment

Xiaoling Sun and Shanshan Li

School of Information Engineering, Institute of Disaster Prevention, Langfang, Hebei, China

## ABSTRACT

Presently, the focus of target detection is shifting towards the integration of information acquired from multiple sensors. When faced with a vast amount of data from various sensors, ensuring data security during transmission and storage in the cloud becomes a primary concern. Data files can be encrypted and stored in the cloud. When using data, the required data files can be returned through ciphertext retrieval, and then searchable encryption technology can be developed. However, the existing searchable encryption algorithms mainly ignore the data explosion problem in a cloud computing environment. The issue of authorised access under cloud computing has yet to be solved uniformly, resulting in a waste of computing power by data users when processing more and more data. Furthermore, to save computing resources, ECS (encrypted cloud storage) may only return a fragment of results in response to a search query, lacking a practical and universal verification mechanism. Therefore, this article proposes a lightweight, fine-grained searchable encryption scheme tailored to the cloud edge computing environment. We generate ciphertext and search trap gates for terminal devices based on bilinear pairs and introduce access policies to restrict ciphertext search permissions, which improves the efficiency of ciphertext generation and retrieval. This scheme allows for encryption and trapdoor calculation generation on auxiliary terminal devices, with complex calculations carried out on edge devices. The resulting method ensures secure data access, fast search in multi-sensor network tracking, and accelerates computing speed while maintaining data security. Ultimately, experimental comparisons and analyses demonstrate that the proposed method improves data retrieval efficiency by approximately 62%, reduces the storage overhead of the public key, ciphertext index, and verifiable searchable ciphertext by half, and effectively mitigates delays in data transmission and computation processes.

Corresponding author
Xiaoling Sun,
sunxiaoling@cidp.edu.cn

# INTRODUCTION

In recent years, target tracking, robotics, manufacturing, and other industries have increasingly employed information from multiple sensors or data sources for analysis. Compared to single-sensor systems, multi-sensor network monitoring systems can filter out irrelevant information by fusing detection data from various sensors, ultimately

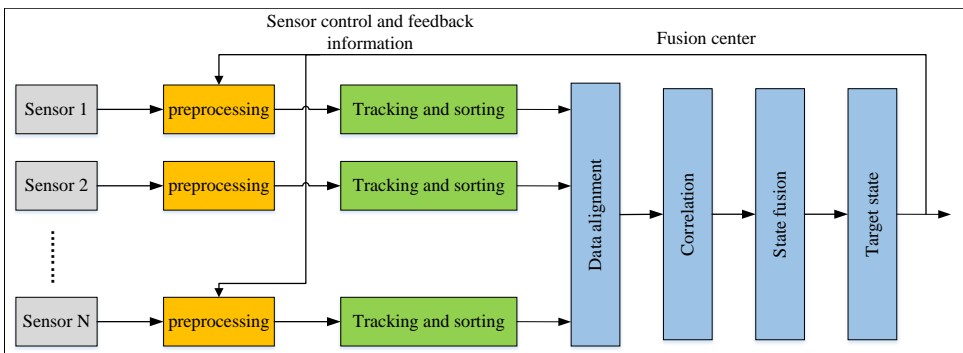

**Figure 1 Distributed fusion tracking system.** As illustrated in the figure, in distributed fusion tracking systems, local tracks are automatically generated and processed on each sensor during data measurement.

leading to optimal target estimation and evaluation. As illustrated in Fig. 1, local tracks are automatically generated and processed on each sensor during data measurement in distributed fusion tracking systems. These sensors subsequently transmit their respective track information to the fusion centre for track correlation and fusion (*Lin et al., 2021*). Each sensor and the fusion centre operate independently in this process with their filters. As a result, the distributed fusion structure provides more excellent reliability compared to centralised fusion structures. Tracking tasks can still be completed even when local sensors, fusion centres, or other potential issues are eliminated. Therefore, within a distributed environment, each sensor represents an intelligent resource that can operate as a local fusion centre for decision-making.

Cloud computing (*Tanweer, 2020*) as a form of distributed computing has been applied across various fields on the Internet, providing expanded opportunities for managing vast and complex data. In multi-sensor networking, cloud computing technology fuses the measurement data from multiple sensors within the cloud storage space, ultimately enabling data observation and tracking. However, storing data in the cloud space also raises security concerns. In the era of big data, sensitive data is ubiquitous, and any potential data leaks can result in substantial losses.

The security issues of cloud storage centres are primarily categorised as external and internal security problems. External security problems refer to hackers who illicitly invade the data source from outside and obtain data. In contrast, internal security problems are primarily caused by the unauthorised use of data by the cloud storage centre or internal personnel who do not have corresponding permissions (*Prajapati & Shah, 2022*). To address these challenges, the primary approach involves encrypting the data before uploading it to the cloud computing centre to guarantee data confidentiality (*Kamara & Lauter, 2020*). However, storing all data in an encrypted format on the cloud makes it necessary to download all the encrypted data locally to search for specific information, significantly consuming computing and communication resources. In 2000, *Song, Wagner & Perrig (2000)* introduced searchable encryption technology to allow searching while maintaining encryption. The searchable encryption algorithm enables encrypted data

storage and retrieval functions, allowing users to search for required data using keywords without exposing the plaintext content of the data stored in the cloud storage centre. The cloud storage centre returns relevant data to users, enhancing search efficiency and ensuring network data transmission security.

The fused structure based on the searchable encryption algorithm (*Ma et al., 2020*) encrypts the local fusion information of all single sites and then transmits it to the cloud. In this way, measurement information is not processed in the communication process, thereby avoiding any loss of information. The cloud fusion system design provides modularization and scalability convenience, allowing for the rapid use of additional sensors. Therefore, in the context of cloud computing, there are still challenges to be addressed in implementing fine-grained multi-sensor networking and keyword tracking while leveraging the benefits of cloud edge computing and searchable encryption technology. Therefore, this article proposes a lightweight, fine-grained searchable encryption scheme to achieve secure access and fast data search in multi-sensor networking tracking in the cloud computing environment. Complex calculations are placed on edge devices by encrypting and calculating trap gates through auxiliary terminal devices to improve security and performance.

## RELATED WORK

In most scenarios, a single sensor can only capture a limited scope of information within its immediate vicinity. This inherent limitation necessitates the utilisation of multi-sensor networked cooperative tracking to obtain complementary tracking information across various regions, rendering multi-sensor systems increasingly significant across both military and civilian fields. In a multi-sensor networked cooperative tracking system, the composite tracking information from numerous sensors is captured globally through amalgamation, which results in a combined tracking effect through the implementation of information fusion technology, thereby achieving superior performance. The quality of the target tracking algorithm and the effectiveness of the sensor management algorithm decisively determine the performance of the multi-sensor networked cooperative tracking system. The system's optimal performance is accomplished by an excellent target-tracking algorithm paired with sensors optimised by a sensor management algorithm, thereby enhancing the system's tracking proficiency and resource utilisation with the least possible sensor resources (*Hill & Larson, 2023*).

In recent decades, the rapid advancement of sensor technology has resulted in many software-controlled, easily modifiable sensor devices that can execute various sensing functions (*Kreucher, Hero & Kastella, 2005*). By integrating random set theory, *Kreucher, Hero & Kastella (2005)* broadened the scope of the sensor management algorithm based on information theory. In 2005, *Tmazirte et al. (2013)* proposed a novel approach that maximises the alpha-Rényi information gain measurement by utilising sensors to optimise the likelihood of accurate target location after the following measure. *He, Shin & Tsourdos (2018)* developed a new method to process various fields of view within the generalised covariance cross-fusion framework, enabling centralised and distributed point-to-point

sensor networks to perform multi-target tracking. *Yi, Li & Battistelli (2020)*, a Gaussian mixture was employed to achieve tracking on a single sensor with six obstacles with different fields of view and a random line of sight. Although these multi-sensor target-tracking approaches were executed in either the fusion centre or the sensors themselves, they suffered from data storage and computing redundancy issues, ultimately impeding effective target state acquisition.

In the 5G era (*Jorge et al., 2020*), the widespread expansion of the Internet of Things has led to a substantial proliferation of Internet of Things (IoT) terminal devices (*Singh, Singh & Goyal, 2021*), resulting in a significant upswing of sensor data. The pre-established data-centric storage mode is no longer viable for massive data processing, requiring computing equipment to have an enhanced real-time and security processing mode to cope with this unprecedented demand. Thus, the concept of cloud computing was conceived (*Lu et al., 2021*). The widespread adoption of mobile cloud computing has stimulated the emergence of a broad range of intelligent applications, which has fueled the escalating demand for various data types available in the cloud. In this context, sensor data represents the centrepiece of such applications, enabling the conflation of different sensor data to enable comprehensive analysis. An extensive ETL and filtering process can expedite data fusion quickly and efficiently, minimising sensor data processing overhead.

With the rapid proliferation of cloud computing, an ever-growing number of users prefer to store their data in the cloud. However, storing and processing data on cloud servers can engender problems of data leakage and destruction during data transmission. Consequently, searchable encryption algorithms can support data retrieval and calculation in ciphertext storage. Building upon *Song, Wagner & Perrig (2000)*, *Hur & Noh (2010)* proposed a ciphertext policy attribute-based searchable encryption algorithm, which facilitates the implementation of access control policies with a user recall function and a practical attribute-based search function. To improve efficiency, *Zheng, Xu & Ateniese (2015)* has implemented fine-grained access features in searchable encryption and utilised the tree structure to assign special access permissions to each attribute.

Additionally, *Cheng et al. (2015)* proposes a verifiable scheme to prevent malicious servers through security-based indiscernibility confusion. Nonetheless, this scheme does not offer forward privacy. To support forward privacy and the single-owner, multi-user model, *Zhu et al. (2018)* pioneered a general verifiable symmetric searchable encryption scheme, which also supports the verifiability of any symmetric searchable encryption scheme. *Yang et al. (2019)* proposes an effective searchable encryption scheme that can verify computation while preserving output privacy. The scheme achieves blind verifiability, enabling the verifier to authenticate the integrity of the results without any knowledge. Finally, *Shen et al. (2018)* developed a searchable and verifiable solution to support big data applications. The scheme employs the cube data structure, which is convenient for storage and access but cannot guarantee data sharing security. The document set encryption scheme can save storage space, ciphertext encryption, and decryption time. Compared with the static solution, the dynamic solution that supports ciphertext data updates (adding or deleting) is more practical. In *Guo et al. (2017)*, the OMAP structure is used to hide the access address of the server ciphertext dictionary, which implements a secure

symmetric searchable encryption scheme that supports dynamic updating. However, the high communication cost of the OMAP structure reduces the practicality of the scheme. *He et al. (2020)* designed a two-level index chain structure to achieve a fixed size of client storage space overhead, but this scheme is unsuitable for multi-sensor networking tracking environments.

# LIGHTWEIGHT SEARCHABLE ENCRYPTION FOR MULTI-SENSOR NETWORKING TRACKING IN CLOUD EDGE COMPUTING

The symbols used in this scheme are shown in Table 1.

The basic concepts involved in this scheme are defined as follows:

Data owner (DO): Encrypt the text, establish an encryption index through the key and selected keyword set, and then send it to the cloud computing centre for storage.

Cloud server (CS): With almost unlimited computing and storage capacity, the cloud computing centre can undertake remote file storage tasks and ciphertext retrieval operations, reducing the computing pressure on sensors.

Data user (DU): terminal data users with limited resources can generate the final searchable encrypted ciphertext with the help of edge nodes and trapdoors to send search queries with the help of edge nodes. The terminal DU needs to decrypt the final ciphertext returned by the edge node.

Edge node (EN): The edge node can generate the final ciphertext and help the data owner and terminal data user output the final trap door. Moreover, EN can partially decrypt the returned ciphertext to reduce computing costs further.

## Lightweight searchable encryption in edge computing

The Scheme implementation flow chart (Fig. 2) shows that each sensor conducts information detection in the multi-target detection environment and transmits target-related information to the cloud server for multi-level closed-loop information tracking. This scheme mainly targets two scenarios. The first one is to re-encrypt the initial public key encryption with keyword search (PEKS) ciphertext generated by the multi-sensor using the public-key searchable encryption technology for the number captured by the sensor. Finally, the generated PEKS ciphertext is stored in the cloud centre. The second scenario begins if a user or organisation wants access to the cloud centre's valuable sensor storage data. The side server generates the PEKS search trap for the required data file. After receiving the trap, the cloud centre searches for the corresponding ciphertext and returns it to the user. The solution must meet this case's data privacy and availability requirements. Therefore, this article designs the searchable encryption algorithm under the participants and process shown in Fig. 2 to realise the sharing of cloud data.

## Initialization and key generation

To solve some technical defects of searchable encryption in the existing traditional cloud computing environment, such as the high time cost when only using terminal devices to generate PEKS ciphertext or trapdoors for search, it is difficult to ensure security when

**Table 1  Scheme symbol list.** The symbols used in this scheme are shown.

| Symbol | Description |
| --- | --- |
| $MSK$ | Master key |
| $PP$ | Public parameter |
| $F$ | Date uploaded to the cloud server |
| $PK_{DU}, SK_{DU}$ | The public and private keys of the data user |
| $PK_{EN}, SK_{EN}$ | The public and private keys of the edge node |
| $w$ | Keyword set |
| $I$ | Index set |
| $Z_p^*$ | The set of positive integers less than $p$ and congruent with $p$ |
| $Z_p$ | A set of integers from 0 to $p-1$ |
| $T_{w'}$ | Search trap |
| $UL$ | User authorization list |
| $x \in_R Z_p^*$ | Choose $x$ from $Z_p^*$ at random |

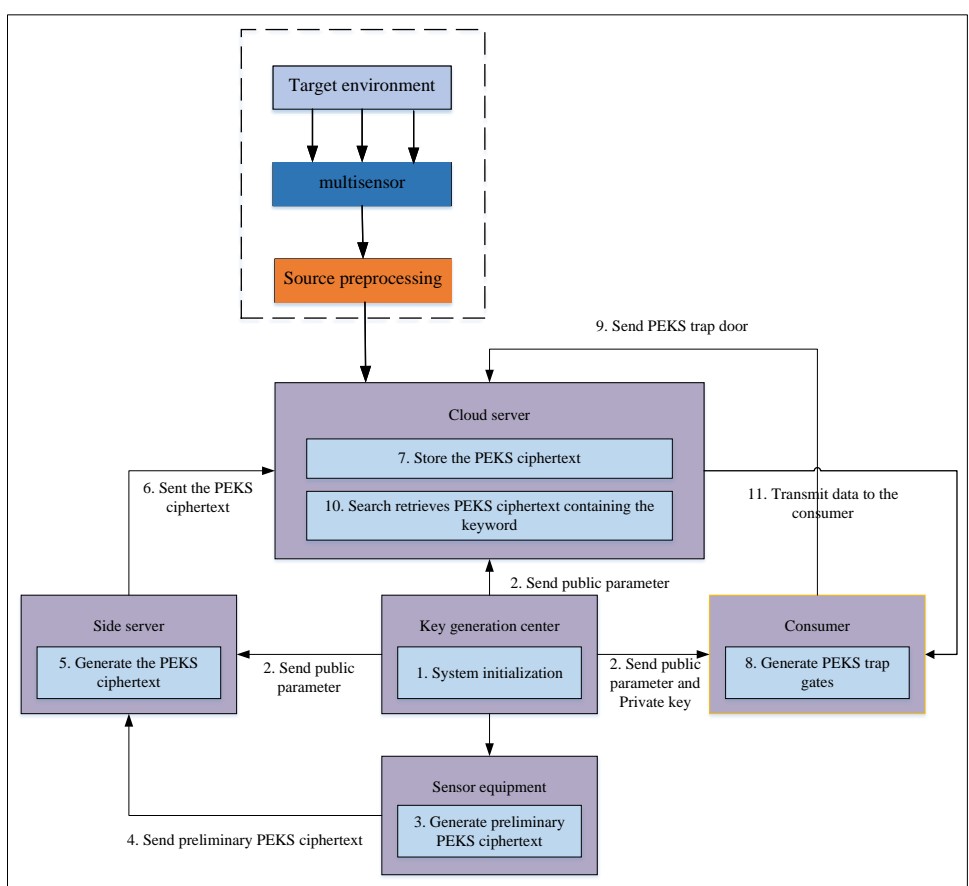

**Figure 2  Program implementation process.** The scheme implementation flow chart shows that, in the multi-target detection environment, each sensor conducts information detection and transmits target-related information to the cloud server for multi-level closed-loop information tracking.

only using edge devices to generate PEKS ciphertext or trapdoors for search. The scheme research in this chapter will use edge servers for the auxiliary calculation to help terminal devices generate PEKS ciphertext and search trapdoors, improve the efficiency of ciphertext generation and retrieval without losing security, and reduce the energy consumption of terminal devices.

System Initialization Setup$(1^k)$: Input the safety parameters given by the system $k$, generate public parameters $pp = (G, G_T, e)$, the algorithm is executed by KGC. Select generator $g_0, g_1 \in G$, 2D vector $x, y \in_R Z_p^2$ and hash functions $H_1 : \{0,1\}* \to_R Z_p^*$; Then for each $att_i \in U (i \in [1,n])$ attribute, randomly selects a non-zero value $t_i \in_R Z_p^*$, Calculate $A_i = g_0^{t_i}, B = g_0^x, Y = e(g_0, g_1)^v$; Finally, the KGC returns the public parameters and master key.

$$
\begin{aligned}
PP &= (pp, g_0, g_1, B, Y, H_1, A_i) \\
MSK &= (x, y, t_i)
\end{aligned}
\tag{1}
$$

Key generation KeyGen$(PP, MSK, S)$: When an edge node joins the system, the KGC first select EN as the intermediary, then select $s \in_R Z_p^*$. To generate a public key $PK_{EN}^* = PK_{EN}^{-s} = e(g_0, g_1)^{-syr}$ and authorized user list (UL) to establish public key relationship $PK_{EN}^*$ with shared data. When each DU of UL is added to the system.

The KGC first select for $a_j, b_j \in_R Z_p^2$, $att_j^* \in S$ and $v, z \in_R Z_p^*$, and then it calculates $K_{j,1} = g_0^{b_j/t_{p1}(j)}$, $K_{j,2} = g_0^{(a_j-b_j)/t_{p1}(j)}$, $K_{j,3} = g_0^{xv/t_{p1}(j)}$, $K_0 = g_0^{y+xv}$, $K_1 = g_0^{to} g_1^2$, $K_2 = g_1^2$, $K_3 = g_0^{y-a}$, finally, the KGC will return $SK_{DU} = (K_0, u, K_{j,1})$, $SK_{EN} = \{K_1, K_2, K_3, r, \{K_{j,2}, K_{j,3}\}\}$ to DU and EN respectively.

## Specific implementation

DO send $\Gamma$ to an EN, and output the file symmetric key $CT_\tau$ through cooperation with the selected EN.

After obtaining $\Gamma$, for each node $w$ in the tree T, select one of the specified EN $d_w$ Degree polynomial $q_w$. set up $L = \{l\}$ as the set of leaf nodes in T. Then EN calculates $C_1 = g_0^\theta, C_2 = g_1^\theta, C_l = g_0^{t_l q_l(0)}$. Finally, EN returns the $CT_\tau$ to DO, as shown in Formular Eq. (2):

$$
CT_\tau = (C_1, C_2, C_{l l \in L})
\tag{2}
$$

Then DO select $h \in_R Z_p^*$, calculate $CT_\tau = C_1 g_0^h$, $C_2' = C_2 g_1^h$. Finally, DO sends the $CT_\tau$ and $CT_\tau^*$ return to CS through the selected EN, where $CT_\tau^*(CCT_\tau, C', C_1, C_2, \{C_l\}_{l \in L})$.

Using DO to generate files uploaded by each sensor $F_\tau$ index of each attribute selected in $d_l \in_R Z_p^*$, calculate $l_0 = Y^s, l_1 = g_0^s, I_1 = g_0^s, I_{l,1} = A_l^{d_1 H_1(W)}$, $W$ represents the subsequent data retrieval is realized through keywords $F_\tau$. Finally, DO will use the above EN to set the final index set $I = \{I_\tau\}$ and tuples $(\{c_\tau\}, C_\tau^*)$ to send to CS, where represented as $I_\tau = (I_0 I_1 (I_{\tau,1} I_{\tau,2l \in L}))$.

For each attribute $att_j^* \in S$, DU outputs $T_{j,1}^l = T_{j,1}^{\lambda/H_1(w')}$, $T_{j,2}^l = K_{j,1}^{\lambda/H_1(w')}$. Finally, the DU will return $T_{w',2} = (T_0, T_1', T_{j,1}', T_{j,2})$ to the selected EN. When receiving $T_{w',2}$, EN calculates $T_0' = T_0 \eta + r, T_{j,2}' = T_{j,2}^\eta$ for each attribute in S, and calculates $T_{DU}^* = (PK_{DU}^*)^\eta$.

Finally, EN will finally trap the door $T_{w'} = \left( T'_0, T'_1, T'_{DU} T'_{j,1}, T'_{j,2} \right)$ and attribute S of DU are sent to CS.

Search function $Search\left(\{c_\tau\}, CT^*_\tau, I, S, T_{w'}, \Gamma\right)$: After obtaining trapdoors, CS first checks whether the data user has access rights according to the attributes of the DU and the established access structure to achieve secure access to data. When it has access rights, CS calculates $I^*_l = I_{l,1} \cdot A_l^{I_{l,2}} = A_l^{s/H(w)}$ and $e(I^*_l, T'_{j,1}), e(I^*_l, T'_{j,2})$ for $att^*_j \in S$. Then, CS checks legality of the trap door submitted $T_{w'}$, The corresponding search results containing the file ciphertext ($c'_\pi$) and file key ciphertext $CT'_\pi$ are returned to EN.

Decryption function $Dec\left( c'_\pi, CT'_\pi, SK_{EN}, SK_{DU}, PP, T, \Gamma \right)$: To decrypt $c'_\pi$, the following recursive algorithm is adopted to obtain the embedded File key $CT'_\pi$ in $k'_\pi$. The specific decryption process involving EN and DU is as follows: First, EN calls the recursive algorithm by considering these two situations.

(1) When leaf node $w$ satisfy $att(w) \in S$, EN calculates $att(w) \in S\phi_w = e(K_{att(w),3}, C_w) = e(g_0, g_0)^{xvq_w(0)}$;

(2) If the node $w$ is a non leaf node, and EN is the node first calculate $\varphi_{w'}$ for Each child node of $w$, that is $w'$, EN calculate $\varphi_w = \prod_{w' \in \phi} \varphi_{w', S'_w(0)} = e(g_0, g_1)^{xvq_w(0)}$, where $i = index(w') = \{index(w') \in S_w\}$.

(3) If S matches the access structure $\Gamma$, EN obtained $\varphi_v = e(g_0, g_1)^{xv\theta}$. After that, EN calculates $M = e(g_0, g_0)^{xv(\theta+h)}$ and $M^* = M/\varphi_v = e(g_0, g_0)^{xvh}$, and then return $\left( c'_\pi, CC'_\pi, C', M^* \right)$ to DU. Finally, DU calculates each file key decrypt ciphertext $k'_\pi$ before each file satisfy $c'_\pi = E_{k'_\pi}(F'_\pi)$, where $CC'_\pi = CC_{p2(\pi)}$.

$$k'_\pi = \frac{CC'_\pi M_*}{e(K_0, C')} = \frac{k'_\pi \cdot e(g_0, g_0)^{yh} e(g_0, g_0)^{xvh}}{e(g_0^{y+xv}, g_0^h)} \tag{3}$$

## SIMULATION RESULT ANALYSIS

This scheme is improved based on *Yang et al. (2019)*. To further illustrate the efficiency improvement of the proposed scheme, the communication overhead, storage overhead, and computing overhead of each stage of the scheme and *Yang et al. (2019)* are compared in 'Efficiency analysis of each stage'. Meanwhile, to horizontally compare the computing performance of the proposed searchable encryption scheme in the cloud environment, the running time simulation of the encryption index, ciphertext generation, and search encryption data of the proposed scheme and *Lu et al. (2021)*, *Zheng, Xu & Ateniese (2015)*, and *Yang et al. (2019)* under different retrieval data, the number of PEKS ciphertext, and the number of encryption indexes are conducted in 'Comparative analysis of simulation experiments'. The polynomial degree selected in the experiment is 100, the modulus N is 1,024, the random number r and the generator s are 1,024 bit, and the hash value of the hash function is 128 bit. Under this parameter, 100 simulation experiments are carried out, and the final experimental result is the average of the 100 operation results.

**Table 2** **Comparison with the storage overhead in *Yang et al. (2019)*.** The stored data generated after the execution of the two algorithms is shown. It can be seen from table that the space occupied by the public parameters generated in this scheme is 50 times smaller than the space occupied by the algorithm in *Yang et al. (2019)*, and the space occupied by encrypted index and PEKS ciphertext is also half of the space occupied by the algorithm in *Yang et al. (2019)*.

|  | Our scheme | Reference (*Yang et al., 2019*) |
|---|---|---|
| Public key | 2,048 bit | 104,448 bit |
| Encrypted index | 1,024 bit | 2,304 bit |
| The private key of DU | 1,024 bit | – |
| The private key of EN | 2,048 bit | – |
| PEKS ciphertext | 2,048 bit | 4,096 bit |

## Efficiency analysis of each stage

The stored data generated after the execution of the two algorithms is shown in Table 2. It can be seen from Table 2 that the space occupied by the public parameters generated in this scheme is 50 times smaller than the space occupied by the algorithm in *Yang et al. (2019)*. The space occupied by the encrypted index and PEKS ciphertext is also half of the space occupied by the algorithm in *Yang et al. (2019)*. Because sensors and data users need to encrypt data transmission and ciphertext retrieval, the private key cost is relatively large. Still, the private key only needs to be saved and does not need secondary transmission on the communication channel. Therefore, compared with the algorithm in *Yang et al. (2019)*, the space cost of this scheme is much lower. The simulation experiment in Table 3 takes 500 data as an example and compares the computational overhead obtained from the experiment with the encrypted index, generated PEKS ciphertext, and keyword search as variables. The overhead at each stage has been reduced, realising a dual reduction in space and computational overhead.

## Comparative analysis of simulation experiments

Select multiple groups of data to perform the encryption index generation experiment on this scheme and the schemes in *Lu et al. (2021)*, *Zheng, Xu & Ateniese (2015)* and *Yang et al. (2019)*, and the results are obtained in Fig. 3. The algorithm for generating an encrypted index in the algorithm in *Yang et al. (2019)* is proportional to the polynomial degree, the number of keys. The number of keywords selected in the experiment in Fig. 3 is 100. The data in Fig. 3 shows that the encryption index generation in this scheme is faster than in the other three schemes.

Then, this article selects 250, 500 and 1,000 keywords for ciphertext generation experiments and obtains the data in Fig. 4. Whether in this scheme or the other three schemes, the running time is linear with the number of keywords. However, in this scheme's PEKS ciphertext generation process, the part with significant computation is placed on the edge server with greater computing power, so the speed is much faster.

This article selects multiple groups of data items to conduct experiments on this scheme's search encryption data algorithm and the three methods and obtains the data in Fig. 5. Figure 5 shows that the time for searching data increases linearly with the number of

**Table 3   Comparison with the computational efficiency of *Yang et al. (2019)*.** The simulation experiment in Table 2 uses 500 data as an example, and compares the computational overhead obtained from the experiment with the encrypted index, generated PEKS ciphertext, and keyword search as variables.

|  | Our scheme | Reference (*Yang et al., 2019*) |
| --- | --- | --- |
| Key generation | 499 ms | 858 ms |
| Generate encrypted retrieval | 3,635 ms | 6,240 ms |
| Generate the PEKS ciphertext | 8 ms | 16 ms |
| Search encrypted data | 100 ms | 266 ms |

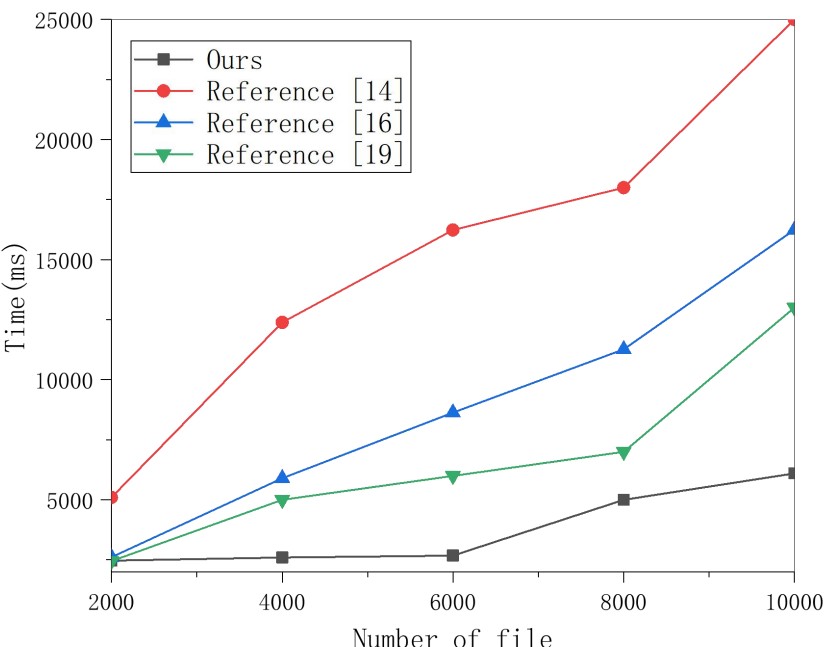

**Figure 3   Generate encrypted index test comparison.** Multiple groups of data were selected to perform the encryption index generation experiment on this scheme and the schemes in *Lu et al. (2021)*, *Zheng, Xu & Ateniese (2015)*, and *Yang et al. (2019)*, and the results are shown in the figure.

encrypted indexes, and the time cost of the three schemes is much higher than that of this scheme. With the increased number of encrypted indexes, the time cost gap is also larger. This is because the three schemes require polynomial operation for keyword matching. In contrast, this scheme only requires one power operation, so the time cost is much lower, which can meet the data retrieval efficiency in multi-sensor networking tracking in the cloud computing background.

## CONCLUSION

To solve the problem of low data processing efficiency in multi-server networking and tracking in a cloud computing environment, this article proposes a lightweight searchable encryption scheme based on edge computing, which adds edge servers between the cloud

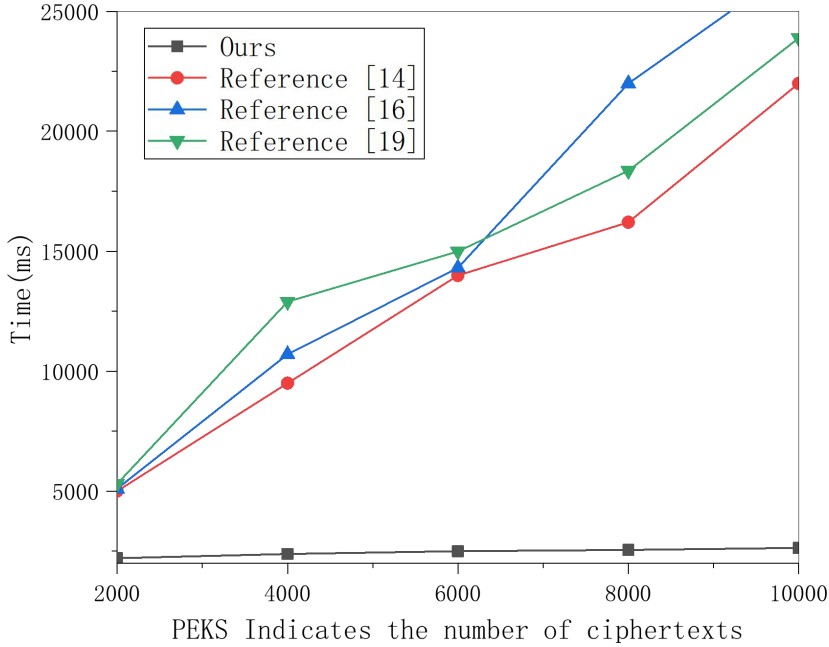

**Figure 4 Test comparison of ciphertext generation algorithm.** A total of 250, 500 and 1,000 keywords were selected for ciphertext generation experiments, and the results are shown in the figure.

computing centre and terminal devices to assist terminal devices in operations with a large amount of computation and reduce the computing pressure on the client side. The terminal device can generate preliminary searchable encrypted ciphertext locally and then send it to the edge server. Afterwards, the edge server can perform bilinear mapping operations with a large amount of computation, generate the final searchable encrypted ciphertext, and transmit it to the cloud computing centre. This can ease the computing pressure on the terminal device while ensuring data security and realising the characteristics of fast data fusion in multi-server networking. Similarly, when searching for searchable encrypted ciphertext, the edge server can also help generate search traps to reduce the energy consumption of terminal devices. Finally, the security of the scheme is compared and analysed. The computational efficiency of the scheme is compared with other searchable encryption schemes through simulation experiments. The experimental results show that the scheme system has higher efficiency and security. However, when the number of sensors in this scheme increases, more node information must be stored on the client side, increasing the terminal data storage overhead. In the future, further research will be carried out under the scenario of reducing the storage overhead on the client side, the scheme's communication overhead, and the networking tracking scenarios of dynamically adding or deleting sensors.

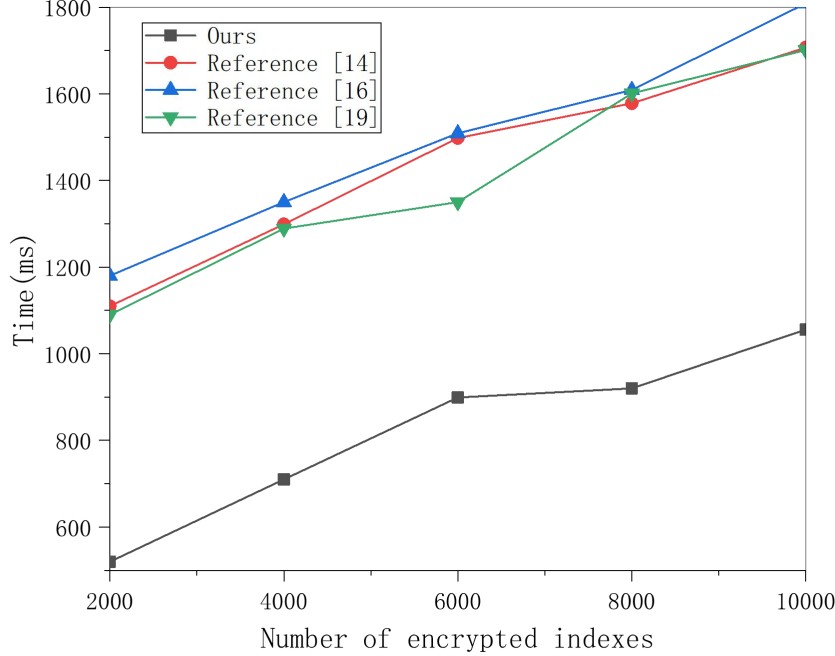

**Figure 5** **Search encrypted data test comparison.** Multiple groups of data items were selected to conduct experiments on the search encryption data algorithm of this scheme and the three other schemes, and the results are shown in the figure.

## Funding
We received funding from the Fundamental Research Funds for the Central Universities (ZY20215152). The funders had no role in study design, data collection and analysis, decision to publish, or preparation of the manuscript.

## Grant Disclosures
The following grant information was disclosed by the authors:
Fundamental Research Funds for the Central Universities: ZY20215152.

## Competing Interests
The authors declare there are no competing interests.

## Author Contributions
- Xiaoling Sun conceived and designed the experiments, performed the experiments, analyzed the data, performed the computation work, prepared figures and/or tables, authored or reviewed drafts of the article, and approved the final draft.
- Shanshan Li conceived and designed the experiments, performed the experiments, analyzed the data, performed the computation work, prepared figures and/or tables, authored or reviewed drafts of the article, and approved the final draft.

## Data Availability

The code is available in the Supplementary File.

The data is available at Stanford and at Zenodo: https://snap.stanford.edu/data/com-LiveJournal.html.

Anonymous. (2023). Network Tracking Dataset [Data set]. Zenodo. https://doi.org/10.5281/zenodo.7855106.

## Supplemental Information

Supplemental information for this article can be found online at http://dx.doi.org/10.7717/peerj-cs.1433#supplemental-information.

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
