# Peer review of "Multi-sensor network tracking research utilizing searchable encryption algorithm in the cloud computing environment"

_PeerJ Computer Science, doi:10.7717/peerj-cs.1433_

## Round 0.1 · original submission · Major Revisions

Please carefully address all the review comments and submit the revised version for the next round of reviews.

Reviewer 1 ·

Basic reporting

1. The Abstract must be self-contained and concisely describe the reason for the work, methodology, results, and conclusions.
2. The background analysis of the abstract is not enough.
3. The contents of the main issues should be elaborated.
4. The introduction of the introduction part coincides with the description of the literature review in the second chapter.
5. In Introduction, what makes this article different from the rest of studies that are available in the literature is not specified.
6. The gap in exiting literature, by arguing what is missing or inadequate in existing solutions and thus your study is necessary is not identified. This needs to be briefly noted in Introduction, and then further elaborated in the Literature Review, with in-depth analysis and substantiation of citations.
7. It is suggested that the author sort out the citations of these two parts again to ensure a certain logic of writing.
8. The language expression of the Conclusion part needs to be optimized.

Experimental design

1. The process or method of data pre-processing should be detailed

Validity of the findings

1. Pay more attention to the introduction of data sets, and if necessary, should add the process or method of data preprocessing.

Additional comments

To solve the problem of low data processing efficiency in multi-server networking tracking in cloud computing environment, a lightweight searchable encryption scheme based on edge computing is proposed in this paper. Edge servers are added between cloud computing center and terminal devices to assist terminal devices to carry out large computation operations. The terminal device can generate preliminary searchable encrypted cipher text locally, and then send it to the edge server. Then the edge server can carry out the bilinear mapping operation with a large amount of computation to generate the final searchable encrypted cipher text, and then transmit it to the cloud computing center. In this way, the computing pressure of the terminal device can be relieved while the data security is guaranteed. Fast data fusion in multi-server networking is implemented.

The authors have presented an interesting idea in the paper. However, I have a few suggestions for the authors to incorporate in the revised draft.

1. The Abstract must be self-contained and concisely describe the reason for the work, methodology, results, and conclusions.
2. The background analysis of the abstract is not enough.
3. The contents of the main issues should be elaborated.
4. The introduction of the introduction part coincides with the description of the literature review in the second chapter.
5. In Introduction, what makes this article different from the rest of studies that are available in the literature is not specified.
6. The gap in exiting literature, by arguing what is missing or inadequate in existing solutions and thus your study is necessary is not identified. This needs to be briefly noted in Introduction, and then further elaborated in the Literature Review, with in-depth analysis and substantiation of citations.
7. It is suggested that the author sort out the citations of these two parts again to ensure a certain logic of writing.
8. Pay more attention to the introduction of data sets, and if necessary, should add the process or method of data preprocessing.
9. The language expression of the Conclusion part needs to be optimized.
10. The content of Conclusion part needs to supplement the limitations and future development direction.
11. The contribution of your study needs to be clearly articulated in Conclusion part. The contribution is not clear and need to be stated.
12. The section needs to include some recommendations for practitioners based on the findings, if appropriate.

Reviewer 2 ·

Basic reporting

In this paper, a lightweight fine-grained searchable encryption scheme based on cloud edge computing is proposed, which is used to encrypt and generate trap gate calculations for auxiliary terminal devices. Complex calculations are placed in edge devices to realize secure access and fast search of data in multi-sensor network tracking, and the computing speed is accelerated under the condition of ensuring data security. Finally, the experimental comparison and analysis of the scheme show that the proposed method can improve the efficiency of data retrieval by about 62%, reduce the storage cost of the public key, ciphertext index, and verifiable and searchable ciphertext by half, and effectively reduce the time delay in the process of data outgoing and calculation. This is a good research paper for the present perspective. You all had a great effort overall. Please follow the following comments for improving the quality of this article.

 For the "Searchable encryption algorithm" mentioned in the title and abstract, the author needs to give a more accurate and professional definition;
 “The cloud fusion system design provides modularization and scalability convenience, allowing for the rapid use of additional sensors.” But I don't see any drawbacks to the current technology. How consistent are the experimental results of this study with the actual application?
 What are the improvements and suggestions for the applied technology?
 The end of introduction should make a supplementary analysis of the research structure.
 In the literature review, the author listed a lot of work in previous years, an objective and clear review is very necessary;
 Please check the Chinese characters in the replacement formula and the redundant space characters in the references.
 There are also some problems in language expression in this paper, which need to be modified.

Experimental design

In this paper, a lightweight fine-grained searchable encryption scheme based on cloud edge computing is proposed, which is used to encrypt and generate trap gate calculations for auxiliary terminal devices. Complex calculations are placed in edge devices to realize secure access and fast search of data in multi-sensor network tracking, and the computing speed is accelerated under the condition of ensuring data security. Finally, the experimental comparison and analysis of the scheme show that the proposed method can improve the efficiency of data retrieval by about 62%, reduce the storage cost of the public key, ciphertext index, and verifiable and searchable ciphertext by half, and effectively reduce the time delay in the process of data outgoing and calculation. This is a good research paper for the present perspective. You all had a great effort overall. Please follow the following comments for improving the quality of this article.

 For the "Searchable encryption algorithm" mentioned in the title and abstract, the author needs to give a more accurate and professional definition;
 “The cloud fusion system design provides modularization and scalability convenience, allowing for the rapid use of additional sensors.” But I don't see any drawbacks to the current technology. How consistent are the experimental results of this study with the actual application?
 What are the improvements and suggestions for the applied technology?
 The end of introduction should make a supplementary analysis of the research structure.
 In the literature review, the author listed a lot of work in previous years, an objective and clear review is very necessary;
 There are also some problems in language expression in this paper, which need to be modified.

Validity of the findings

Please check the Chinese characters in the replacement formula and the redundant space characters in the references.

Additional comments

Already given in basic reporting

---

## Round 0.2 · accepted · Accept

Congratulations, the reviewers are satisfied with the revised version of the manuscript and have recommended the acceptance decision.

Reviewer 1 ·

Basic reporting

The current version of the paper presents an expressive improvement as compared to the previous one.
The authors provided acceptable answers to all my questions and no more issues were detected in the presented manuscript. Therefore, I recommend the acceptance of the paper in its current form.

Minor Edits:
1. All variables should be italicized.
2. All references should be written according to a same format considering the journal referencing style.

Experimental design

The current version of the paper presents an expressive improvement as compared to the previous one.
The authors provided acceptable answers to all my questions and no more issues were detected in the presented manuscript. Therefore, I recommend the acceptance of the paper in its current form.

Validity of the findings

The current version of the paper presents an expressive improvement as compared to the previous one.
The authors provided acceptable answers to all my questions and no more issues were detected in the presented manuscript. Therefore, I recommend the acceptance of the paper in its current form.

Additional comments

The current version of the paper presents an expressive improvement as compared to the previous one.
The authors provided acceptable answers to all my questions and no more issues were detected in the presented manuscript. Therefore, I recommend the acceptance of the paper in its current form.

Minor Edits:
1. All variables should be italicized.
2. All references should be written according to a same format considering the journal referencing style.

Reviewer 2 ·

Basic reporting

The authors incorporated all the required changes.

Experimental design

The authors incorporated all the required changes.

Validity of the findings

The authors incorporated all the required changes.

Additional comments

The paper is acceptable in its current form